# Condition Monitoring of Ball Bearings Based on Machine Learning with Synthetically Generated Data

**DOI:** 10.3390/s22072490

**Published:** 2022-03-24

**Authors:** Matthias Kahr, Gabor Kovács, Markus Loinig, Hubert Brückl

**Affiliations:** 1Department for Integrated Sensor Systems, University for Continuing Education Krems, 2700 Wiener Neustadt, Austria; gabor.kovacs@donau-uni.ac.at (G.K.); hubert.brueckl@donau-uni.ac.at (H.B.); 2Senzoro GmbH, 1030 Wien, Austria; markus@senzoro.com

**Keywords:** condition monitoring, roller bearing, fault detection, machine learning, wavelet transform, simulated training data, 3D multi-body dynamics

## Abstract

Rolling element bearing faults significantly contribute to overall machine failures, which demand different strategies for condition monitoring and failure detection. Recent advancements in machine learning even further expedite the quest to improve accuracy in fault detection for economic purposes by minimizing scheduled maintenance. Challenging tasks, such as the gathering of high quality data to explicitly train an algorithm, still persist and are limited in terms of the availability of historical data. In addition, failure data from measurements are typically valid only for the particular machinery components and their settings. In this study, 3D multi-body simulations of a roller bearing with different faults have been conducted to create a variety of synthetic training data for a deep learning convolutional neural network (CNN) and, hence, to address these challenges. The vibration data from the simulation are superimposed with noise collected from the measurement of a healthy bearing and are subsequently converted into a 2D image via wavelet transformation before being fed into the CNN for training. Measurements of damaged bearings are used to validate the algorithm’s performance.

## 1. Introduction

Reducing economic losses by observing machine degradation before it turns into unexpected downtime is the core task of condition monitoring systems. Prediction based maintenance especially becomes more and more reliable and accurate, thus it may replace schedule based maintenance strategies prospectively. Rolling element bearings are one of the crucial components in mechanical systems, where occurring bearing faults are responsible for up to 40% of all machine failures [1,2,3]. Unfavorable conditions, such as lubrication problems, contamination due to ineffective seals, misalignment and heavy loading, can lead to progressive wear phenomena and induce faults on the bearings’ races or rollers in form of cracks. Thus, material ablation occurs and pits are formed. These local defects produce successive impulses which can be revealed by monitoring the vibration signal of the bearings.

Nowadays, massive collected data, which comprise the health state of a bearing, can be effectively analyzed by using intelligent algorithms including support vector machine (SVM), artificial neural network (ANN) and more recently deep learning architectures. Here, SVM and ANN methods learn patterns based on feature engineering [4,5,6], that is, they rely on expert knowledge, whereas deep learning methods can automatically extract features from raw data [7,8,9]. Comprehensive surveys about traditional and contemporary condition monitoring techniques, including machine learning methods, can be found in [10,11,12,13], respectively.

To successfully train a condition monitoring system based on a machine learning algorithm, a representative quantity of training data is required. This dataset should comprise enough information for the algorithm to learn significant patterns without overfitting the model, which would result in high error rates for unseen data. In general, such datasets can be obtained either experimentally or by simulation [14]. The experimental data-driven approaches are very limited in terms of conducting measurements of prepared bearings with a variety of fault conditions, for example, shape, form and location of a defect, or data are dependent on the collection of in-service data from industry. There is, a priori, no guarantee that a classifier trained with such specific bearing data will accurately classify bearings of the same type but in a different application. Data generated with sophisticated simulation models can overcome this issue with less effort by creating training data, which capture a wider range of operating conditions [15]. The implementation of these data-driven predictive models are of interest for industrial processes such as real-time prediction, sensor calibration and fault detection, as well as process monitoring [16]. Furthermore, training data comprising a combination of simulated and measured data may even enhance the intelligent algorithms.

A common goal in condition monitoring systems is to find fault characteristic frequencies of rotating bearings to assess failure severity and type. Time-frequency analysis methods, such as short-time Fourier transform (STFT), wavelet transform or Hilbert–Huang transform (HHT), are often applied to display fault locations both in time and frequency domains [17]. Those signal processing methods are practicable for deep learning models utilizing 2D convolutional neural networks (CNN), where features are extracted from images during the training process [18].

This work presents a simple 3D simulation model of a ball bearing under different conditions, that is, bearing faults in the form of holes at the inner and outer races. A simplified bearing model is a fast and flexible way to create generic training data and overcomes the issue of a ‘biased’ dataset compared to training data extracted only from measurements. In addition, multiple defective bearings are simulated with limited effort, thus enabling the creation of a ‘bearing condition database’. The data are further superimposed by background noise from measurement of a healthy bearing to facilitate the transfer to a more ‘realistic’ dataset. Next, wavelet transform is applied to create images comprising time and frequency information before a deep-learning CNN is trained on these data. The wavelet transform is used to overcome the limit of a fixed time-frequency resolution in comparison to STFT. To evaluate the CNN’s performance, measurements on real bearings with different health states are conducted and classified by the network. Further, an additional CNN was trained with pure measurement data, to compare performance and classification accuracy with simulated data. Figure 1 summarizes the proposed method.

## 2. Synthetic Data for Training

There are several ways to provide a sufficient amount of data to train a CNN for the classification of ball bearing defects. One way to gather training data is to conduct a variety of measurements with known size and shape of the defects and feed it to the network [19,20]. There are already a variety of data-sets publicly available, featuring bearing damages from measurements. These sets use the classical approach of vibration signals for bearing diagnostics and focus on artificial damages (MFPT [21], MFS [22], Mendeley Data [23]) or real damages (IMS [24]). Another approach relies on the generation of synthetic data, which either can be created analytically by simple functions, that is, with the Dirac delta function and exponential decay for damping, to model the transient force imbalance of a defective bearing, or numerically by solving dynamically dependent equations. Theoretical models to emulate vibration and the influence of various parameters such as loading, shaft frequency and unbalancing effects, multiple defects, cage end play and to some extend the bearing geometry have been developed and consecutively extended [25,26,27,28,29]. Nevertheless, these models are very limited and not all physical phenomena regarding the dynamic behavior of bearings are considered. Hence, for a more detailed investigation, bearing dynamics such as nonlinear spring behavior, variable bearing stiffness, time-dependent excitation of ball bearings, dynamic loadings and so forth are beneficial [30,31,32,33]. These models are based on second order differential equations with a time integration method and differ mainly in the number of DOF (degree of freedom) and the calculus of contacts between the bearing parts (inner ring, rolling elements, outer ring, shaft, etc.).

### 2.1. Simulation Setting

Synthetic bearing vibration data are created with the multibody dynamic modeling software MSCAdams [34]. The bearing components are simplified as rigid bodies and the contact force is numerically evaluated during impacts, which is based on the Hertzian contact theory and considering penalty regularizations. Coulomb friction is enabled and set as greasy steel-greasy steel interaction according to the MSCAdams’s material contact properties. Figure 2 depicts the 3D model and the original bearing. The bearing primary dimensions have been obtained from the manufacturer’s site (skf6200 [35]). However, since important dimensions such as the ball diameter, raceway groove and bearing clearances are not specified by the manufacturer, these parameters have been estimated as follows: the ball diameter is 4.76 mm; the inner and outer ring raceway groove curvature radii are 2.43 mm and 2.48 mm, respectively; the bearing internal clearances (radial play) are set to 20 μm, which corresponds to ISO clearance class C3. To minimize the degree-of-freedom (DOF) and, hence, accelerate computational time, the rollers’ cage is replaced by rigid links to guarantee fixed spacing, but tolerate the rollers’ rotations in 3D space. Additionally, the engine’s intern geometry library RAPID [36,37] was selected, which is a polygon-based interference strategy for contact operations to expedite simulation.

The outer ring is loaded vertically with 100 N and a speed of 3000 rpm (50 Hz) at the inner ring is applied to approximate the bearings real condition (see Section 3). The simulation time step is set to 10 μs with 1 min of simulation time and the outer ring’s acceleration is confined to vertical motion only. A total of 25 simulations, including different inner and outer ring damage sizes in the shape of a hole, have been conducted to provide sufficient amount of training data and to study damage classification dependence of the CNN. The defect sizes are chosen to be 250, 500, 750, and 1000 μm, including a supplementary defect free simulation.

### 2.2. Simulation Results

A few selected simulation results are briefly discussed in this subsection. In general, bearing fatigue yields distinctive fault frequencies, which are related to the bearing’s dimensions. These fundamental defect frequencies are summarized in Equations (Equation 1)–(Equation 4) as ball pass frequency outer race (BPFO), ball pass frequency inner race (BPFI), fundamental train frequency (FTF) and ball spin frequency (BSF) [38,39].
(1)$$BPFO=f_{s}\frac{N_{b}}{2}\left(1-\frac{B_{d}}{P_{d}}\cos\theta \right),$$
(2)$$BPFI=f_{s}\frac{N_{b}}{2}\left(1+\frac{B_{d}}{P_{d}}\cos\theta \right),$$
(3)$$FTF=f_{s}\frac{1}{2}\left(1-\frac{B_{d}}{P_{d}}\cos\theta \right),$$
(4)$$BSF=f_{s}\frac{P_{d}}{2B_{d}}\left(1-{\left(\frac{B_{d}}{P_{d}}\cos\theta \right)}^{2}\right),$$
where $N_{b}$ is the number of balls, $B_{d}$ the ball diameter, $P_{d}$ the pitch diameter, θ the contact angle and $f_{s}$ the shaft frequency.

In Figure 3a, the vibration pattern of a simulated damage-free bearing is shown. Although the bearing is modelled without any defects, a small peak at BPFO and its harmonics are visible. When a roller passes the maximum load zone (top) it slightly elevates the outer ring and vice versa lower the outer ring when exiting the load zone. This modulation is noticeable at the simulation, due to the direct monitoring of the outer ring’s acceleration and the penalty settings for impacts between the bearing’s components. Regular vibration patterns are observed for a faulty bearing with single outer ring damage in Figure 3b, presenting pronounced peaks according to the bearing’s BPFO. The spectrum of a single inner race defect shows several harmonic peaks of BPFI and is also characterized by sidebands, modulated with the shaft frequency (Figure 3c). These sidebands are created due to amplitude modulation of the rotating inner ring with the moving defect. In detail, the defect on the inner race moves in and out of the bearing load zone causing a variation of contact force between the rollers and the raceway. Finally, Figure 3d depicts the vibration pattern of a damaged bearing with both inner and outer ring defects. Note that only the first three harmonics of BPFO and BPFI are marked with colored arrows in the figure. The detected fundamental frequencies BPFO and BPFI conform with the calculated frequencies BPFO=152.4 Hz and BPFI=247.6 Hz from Equations (Equation 1) and (Equation 2), with the parameters set to $N_{b}=8$, θ=0, $f_{s}=50$ Hz, $B_{d}=4.76$ mm and $P_{d}=20$ mm.

## 3. Measurement

Ball bearings have a low starting friction and are mainly used where stop-and-go processes occur, such as in the automotive industry. Nevertheless, bearings experience fatigue, wear, corrosion and deformation enforced by operation in harsh conditions, misalignment, insufficient lubrication, contamination and so forth. In this work, the standard ball bearing skf6200 [35] is thoroughly studied as a healthy bearing and with artificial damage to the outer ring. Carbide micro drills have been used to produce holes through the high-strength outer ring of the bearing. Due to difficulties in drilling, only fault sizes of 500, 700, 900 and 1200 μm could be achieved, which mainly cover the range of the simulated fault sizes. In addition, the small size of the bearing inner ring made it impossible to produce defects by drilling there. Figure 4 depicts a bearing with 0.7 mm single outer ring damage.

The custom made test rig to acquire the bearing data from measurements is depicted in Figure 5. Here, a 2 kW induction motor drives the test bearing at a constant speed of 3000 rpm. The outer ring of the bearing is clamped into the test bearing housing and equipped with analog MEMS accelerometers (ADXL1002, Analog Devices, Norwood, MA, USA). The analog–digital converter (ADC) samples continuously at 100 kHz and enables the recording of up to six analog sensors. An external load of 100 N is radially applied to the housing unit. Thus, the load zone is developed at the bearing’s top part. The bearing’s fault is also located on the top, where the contact force between ball and ring is at a maximum. Further, all bearings were lubricated with grease.

### Measurement Results

The measurement results for a healthy bearing and bearings with different outer ring defect sizes are shown in Figure 6. A defective bearing is characterized by low-frequency components generated by the impact of the rolling element with a fault and high-frequency components which represent the natural frequencies of the rolling element, including background noise from machinery. When the signal is dominated by noise, conventional methods such as the Fast Fourier Transform (FFT) are inappropriate for exposing characteristic failure frequencies. Therefore, signal demodulation (enveloping) based on the Hilbert transform is applied to isolate those components and, thus, emphasize the fault signature [40]. This technique is extensively used in machinery fault diagnosis.

Note that the envelope technique is only used to express the fault frequencies more clearly, shown in Figure 6. The spectra in Figure 6b–d depict an increase in amplitude at the fault frequency BPFO, corresponding to the damage size.

## 4. Convolutional Neural Network

A convolutional neural network (CNN) performs feature extraction and classification for images within a single network-architecture. Here, the automated learning and optimization of filters during the training step outperform manual feature engineering, which is very time consuming for complex data, and further requires no prior expert knowledge [41]. In the following subsections, data preprocessing steps, wavelet transformation to convert the vibration data into 2D time-frequency images, and the settings of the CNN model using these images as training data are explained in more detail.

### 4.1. Data Preprocessing

The bearing fault data from the simulation are very simplified in terms of lubrication, background noise and the calculation of the contact force during collision between the bearing parts, which is the nature of a rigid body model. Further, the direct monitoring of the outer ring during the simulation with neglected damping from housing, the sensor’s transfer function, electronic losses and so forth, yields an imbalance of the simulated and measurement-obtained acceleration data. The amplitude of a measured defective bearing (Figure 6d) is roughly a factor of 100 lower compared to the simulated defect (Figure 3d). Therefore, the synthetic time-series was normalized and superimposed to a dataset of a healthy bearing obtained by measurement in order to transform it into similar amplitude levels. This augmentation also introduces natural noise from the measurement, which intends to generate a more ‘realistic’ synthetic data. An optional moving average filter smooths the peaks even further. Figure 7 depicts the data preprocessing steps.

### 4.2. Wavelet Transformation and Image Compilation

There are several possibilities for converting the one-dimensional time series of the vibration data into a two-dimensional form, which can be processed by the CNN: data stacking, short time Fourier transform (STFT) and wavelet transform. In data stacking, one-dimensional data are stacked row by row to form a 2D input matrix. This is accomplished by equally sliced parts from the time series [42], data obtained by multiple sensors and fused [43], or collecting features from time or (and) frequency domains by applying statistical methods [44,45].

The STFT and the wavelet transform enable the analysis of non-stationary vibration signals in the time and frequency domains simultaneously, thus representing the data in two-dimensions, that is, a spectrogram. The STFT performs a Discrete Fourier Transform on uniform segments of the time series due to a fixed window size. This limits the analysis of signals that contain fast transient features, such as the impact of a ball passing over a local defect. To overcome this limit, the wavelet transform is introduced, which, through translation and dilation of the wavelet, offers both good frequency resolution and poor time resolution at low frequencies and vice versa for high frequencies [40]:(5)$$\psi_{s,\tau }\left(t\right)=\frac{1}{\sqrt{s}}\psi \left(\frac{t-\tau }{s}\right),$$
where *s* and τ are the scale (dilation) and translation parameters, respectively, and ψ(t) the wavelet function. A continuous wavelet transform (CWT) and, thus, a time-scaled analysis of a signal x(t) with the analysing wavelet ψ(t), can be described as convolution:(6)$$\mathrm{CWT}\left(s,\tau \right)=\frac{1}{\sqrt{s}}\int_{-\infty }^{\infty }x\left(t\right)\psi^{*}\left(\frac{t-\tau }{s}\right)dt,$$
where $\psi^{*}$ is the complex conjugate of ψ. There are different types of wavelets such as Morlet, Daubechie, Haar, Gaussian and so forth. Their selection depends on the feature to be extracted from a signal [46]. Some approaches rely on a trial-and-error technique to find the best wavelet for the signal analysis [47,48]. In this work, the Morlet wavelet is used which is defined as the product of a complex exponential function and the Gaussian function, hence, similar to an impact generated transient signal [40]:(7)$$\psi \left(t\right)=\frac{\sigma }{\sqrt{\pi }}e^{-t^{2}\sigma^{2}}e^{i\omega_{0}t},$$
where $\omega_{0}$ is the centre frequency of the window, and σ balances the decay rate of the wavelet. The Morlet wavelet signature matches very well the transient vibration signals of a defective ball bearing with holes or pittings in the race way [49].

Figure 8 depicts the power spectral density (PSD) of simulation and measurement data after wavelet transform. Note, the simulation data consist of sharper pronounced peaks, which result in higher concentrated power at high frequencies. Further, the harmonics are more confined at the frequency level, compared to the broadened appearance of the frequency component of the measurement. The arrows on the top mark the occurring high frequency pattern when the balls overrun the outer ring defect. Here, Δt=6.6 ms corresponds with BPFO. The first three harmonics of BPFO are indicated as red lines on the left. Note that the frequency scale is set to the binary logarithm, to increase the visibility of the low frequency components.

The PSD images of the simulated data and measurement are then fed to the CNN for training and fault prediction.

### 4.3. CNN-Network Architecture

The CNN is designed to extract features from the wavelet PSD images of the simulated bearings. These generated feature maps and, thus, learned patterns are used to classify real bearing data from measurement. A total of 12 different output classes (labels) are available from simulation, which include the healthy state, single inner ring defect, single outer ring defect and both the inner and outer ring defect, all with different hole diameter. These mutually non-exclusive classes (labels) require a multi-label classification task, hence, the probability of each class will be predicted. The aim is to train the network regarding the nature of the defect and the defect size.

The input shape of the wavelet transformed data is 4096 × 160 pixels (time × frequency-scales). This is the equivalent of a data snippet of 40.96 ms (sampled at 100 kHz), and, hence, contains at least six ‘bumps’ of the balls rolling over an outer ring defect (BPFO=151 Hz). The 160 frequency-scales are distributed up to 11 kHz, which corresponds to the linear frequency range of the MEMS accelerometer.

To train the network, a set of 18,000 training and 6000 validation data images from simulation is used. These images are processed by four convolution and subsequently max-pooling layers. The number of features learned during convolution is increased at every convolution layer and complexity is reduced by the max-pooling layers. The activation for the hidden layers is defined by a rectified linear activation function (ReLu). The transition between feature extraction and the classification task occurs by ‘flattening’ the network, thus creating a fully connected (FC) ANN. The last layer uses the sigmoid activation to finally predict the probability of the classes. Backpropagation is used to update internally the weight and offset parameters of the CNN with the aim of reducing output error. ‘Adam’ was selected as the optimizer with a learning rate of 0.001, which are the standard settings in Keras [50]. The ’BinaryCrossentropy’ class was chosen as a loss function and the network metric was set as ‘accuracy’ to track the model’s performance (how often predictions are equal labels). The best performance achieved with this setting is 98% with the training data. However, this work studies the performance of a CNN trained with synthetically generated data for the classification of real data obtained by measurements. A detailed description and theoretical background of the architecture of a CNN can be found in [17,18]. Figure 9 summarizes the network architecture used in this study.

## 5. Results

Two CNNs have been designed; one trained with simulation data, which is normalized and superimposed with data from a measured healthy bearing, and the other trained with pure measurement data. A comparison of both networks demonstrates the networks’ classification accuracy in dependence on the trained datasets. The network trained with simulation data is constructed according to the description in Section 4.3. The other network has the same number of layers (network depth) and features, but only consists of five output classes. These classes describe the five bearing conditions from measurements, that is, one healthy bearing and four bearings with a single outer ring damage ranging from 0.5–1.2 mm. All figures show the prediction of the bearing condition during the long-term measurements of several days.

### 5.1. CNN-Trained with Simulation Data

The time dependence of the prediction of the network trained with simulation data is shown in Figure 10. The prediction in percentage gives the probability of the existence of a certain hole size in the inner or outer ring. It essentially stays constant in most cases. In some cases, the behavior indicates changes of the fault (e.g., Figure 10i).The prediction of the outer ring condition of a healthy bearing (f) and bearings with severe faults of 0.9 mm (i) and 1.2 mm (j) hole sizes are very well classified. Note that the fault sizes from measurement slightly differ from the simulated faults which form the output classes. The bearing with an outer ring defect of 0.5 mm (g) is not accurately predicted and is characterized with lower values (significance) of a healthy bearing state. The condition of a bearing with 0.7 mm fault size (h) is indicated with about 40% probability for the presence of a 0.75 mm defect size. The probability of other defects is much lower.

Since no inner ring damage is introduced, the inner ring’s race condition should be classified as a healthy state. The prediction partly works for the healthy bearing (a) and the bearing with 0.5 mm fault size (b). There is a misclassification with almost 0% (<25%) probability of all defect classes for the cases (c), (d) and (e).

The reproducibility of the classification is investigated by additional measurements, which have been conducted on healthy bearings and bearings with 0.9 mm and 1.2 mm outer ring defects (Figure 11). The network delivers a highly confident classification of the healthy bearings and bearings with 1.2 mm defect size, similar to that shown in Figure 10. There is a run-in phase visible for the ball bearing with 0.9 mm fault size (sample B, Figure 11c) until the network’s prediction stabilizes at 50%. This also reduces the appearance of a clear fault signature, which can be measured by the vibration sensors. Sample C with a 0.9 mm fault size (Figure 11d) is poorly classified (all faults < 25%). Here, a possible deviation from drilling perfection could have an impact on the classification accuracy.

### 5.2. CNN-Trained with Measurement Data

Figure 12 depicts the prediction of a network trained with 4096 × 160 images of wavelet transformed measurement data. For training, 2000 samples of each class are extracted from the measurement of the given bearing samples. This dataset has been isolated for the classification task. Figure 12b) shows a longer run-in phase until the trained network is highly confident of predicting an outer ring fault size of 0.5 mm.

Classifications of the additional measured bearings are shown in Figure 13. The predicted condition of the healthy bearings and bearings with 1.2 mm fault are highly confident, whereas the bearing with 0.9mm fault size is characterized with a run-in phase and also shows a small probability of a present 0.7 mm defect size (Figure 13c). This might be due to variation of the drill angle, deviation of the drill location, drill residues such as burr or smoulder, variation of the load angle (contact force between balls and ring) and the amount of lubrication.

## 6. Discussion and Outlook

The network trained with simulation data, which are superimposed to the vibration data of a healthy bearing, can clearly distinguish between healthy and defective bearings.

While the prediction of outer ring defects works quite well, the correct classification of the inner ring condition fails for most of the cases or indicates wrong defect sizes and low probabilities of different defects (e.g., Figure 10c–e). The reasons for this are unclear. Possible failure sources are described in the following.

Some aspects have to be considered which affect the classification performance. First, the simplified rigid body model of the bearing cannot fully describe the fault patterns compared to a bearing under real condition. A more sophisticated model, which accounts, for example, for lubrication or waviness of the bearing races, would improve the classification process but also increase model complexity and computing time. A compromise between model accuracy and computational time has to be found. Second, a neglected damping mechanism in the measurement setup such as the bearing housing and the sensor mounting, as well as surrounding noise from machinery, also affects the classification accuracy. This is partly eliminated by normalizing and superimposing simulation data with real vibration data of healthy bearings to capture some conditions from the setup. Third, the size deviations of simulated and real defects also influence the network predictability. The shape and placement of a fault within a 3D bearing model can be easily controlled, whereas drilling through hardened steel and occurring burr cannot guarantee an exact replica as a designed fault in a simulation. Fourth, there are many possibilities for how a CNN can be designed, in terms of network depth, parameter fine tuning, activation functions, the selected optimizer, kernel size, input shapes, dropout. and so forth. A more detailed investigation of the network may also improve the classification outcome.

The network trained with data obtained from measurement performed better than that obtained from simulation. This is not surprising, due to the strong correlation between training data and data to be classified which comprises the bearings conditions. Figure 12 especially shows good results, where the training data are extracted from these measurements. Here, the classification performance of the correct classes is 92.5 % on average.

Figure 14 compares the average prediction performance of the CNN trained with simulation data and trained with measurement data in a box-plot graph. For the network trained with simulation data, the mean values of the predicted bearing condition over time is selected from the class which most likely represents the real fault (e.g., defect size class 1 mm for bearing with 0.9 mm and 1.2 mm outer ring fault). The large span of the data is due to the poor prediction accuracy of the small outer ring defect of 0.5 mm (Figure 10g) and the deviation of sample C with 0.9 mm defect (Figure 11d). The mean prediction performance for simulation is 60.5 % while the mean value for the network trained with measurement data is 88.3 %. The outlier in the measurement data represents sample B with a defect size of 0.9 mm.

In conclusion, data from a simulation of a simplified bearing model is a fast and flexible way to generate a variety of different bearing fault conditions, which lay the basis for CNN interpretation of real bearings for predictive maintenance. In a prospective way, a ’bearing condition database’ may be created from simulation, which comprises different bearing faults, such as fault shapes, number of faults, fault size, fault location, and so forth. This database could be used to train algorithms for a specific bearing to monitor its condition and the progress of fatigue during operation. Thus, both costs and time are reduced compared to manually preparing various bearings with artificial defects and then measuring it to obtain training data. This simulation method would be especially beneficial for the condition monitoring of large and expensive bearings, where spare parts are rare and manual bearing preparation is costly.

## Figures and Tables

**Figure 1 sensors-22-02490-f001:**
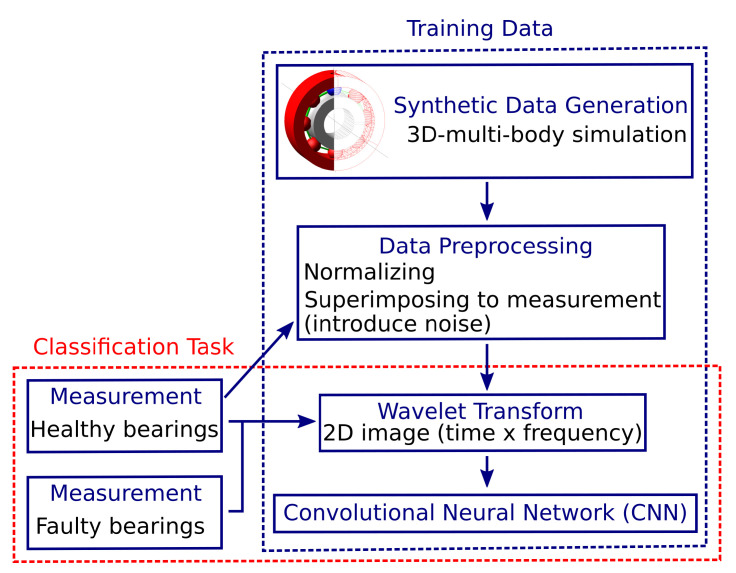
Outline of the proposed method. The CNN is trained with synthetically generated vibration data from a 3D ball bearing model. Noise is introduced from measurement data of a healthy bearing. For comparison, a second CNN was trained with measurement data.

**Figure 2 sensors-22-02490-f002:**
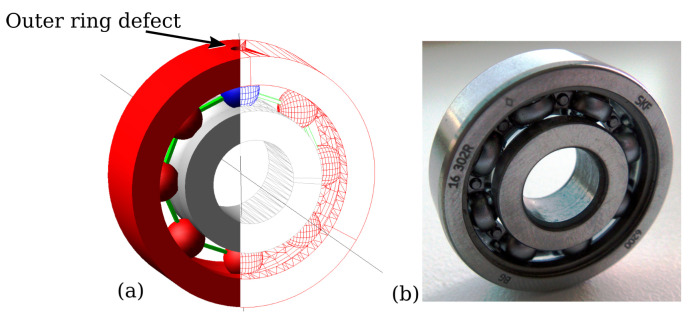
For the generation of synthetic vibration data, a 3D model of the bearing of interest (**a**) was developed. A single outer ring hole defect is exemplarily shown and placed in the top center of the bearing, where the contact force between roller and ring is at the maximum due to vertical load. The original bearing skf6200 is depicted in (**b**).

**Figure 3 sensors-22-02490-f003:**
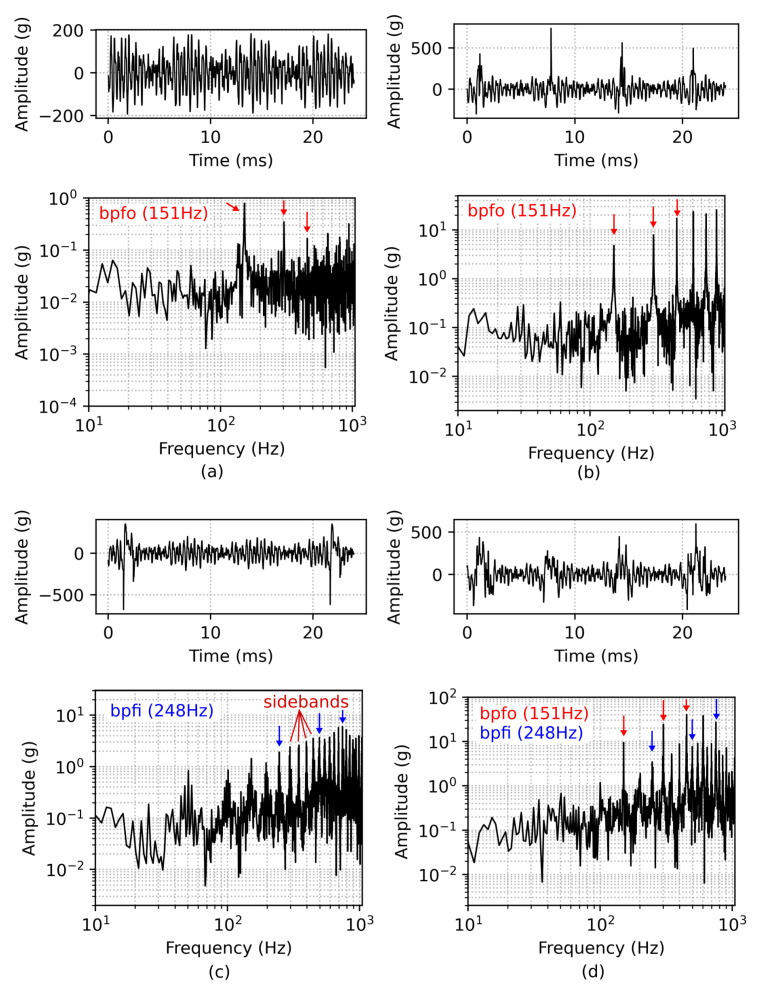
Time snippet of vibration data and corresponding FFT from simulation, depicted on top and bottom of each subplot, respectively, for different simulation settings. Time series and FFT of a damage-free bearing is shown in (**a**). The spectrum of the bearing with a single outer ring fault of 500 μm and a single inner ring fault of 750 μm is characterized by the presence of harmonic peaks, shown in (**b**,**c**), respectively. Figure (**d**) depicts vibration and spectrum of a simulated bearing with both (1000 μm) outer and inner ring damages.

**Figure 4 sensors-22-02490-f004:**
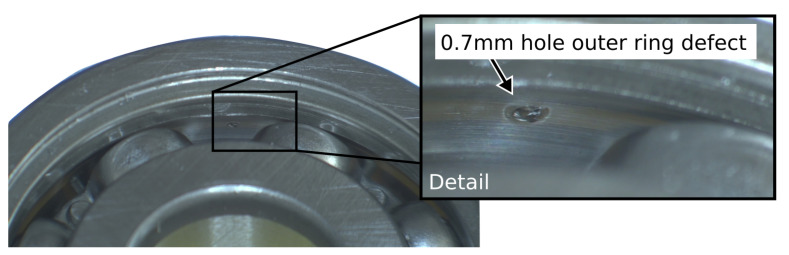
Example of a prepared bearing with single outer ring damage.

**Figure 5 sensors-22-02490-f005:**
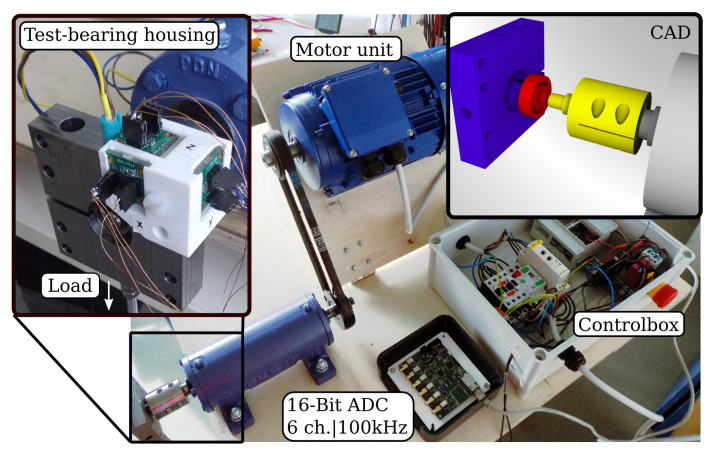
Measurement setup for the condition monitoring of ball bearings. The setup consists of a control box, including a motor protection unit, motor soft starter and a raspberry pi minicomputer which controls the measurement process and the data flow. The exploded CAD view on the top right corner shows the test-bearing housing, a ball bearing unit and the shaft adapter coloured in blue, red and yellow, respectively. The inset on the top left depicts the assembled test-bearing housing with attached temperature and MEMS accelerometers.

**Figure 6 sensors-22-02490-f006:**
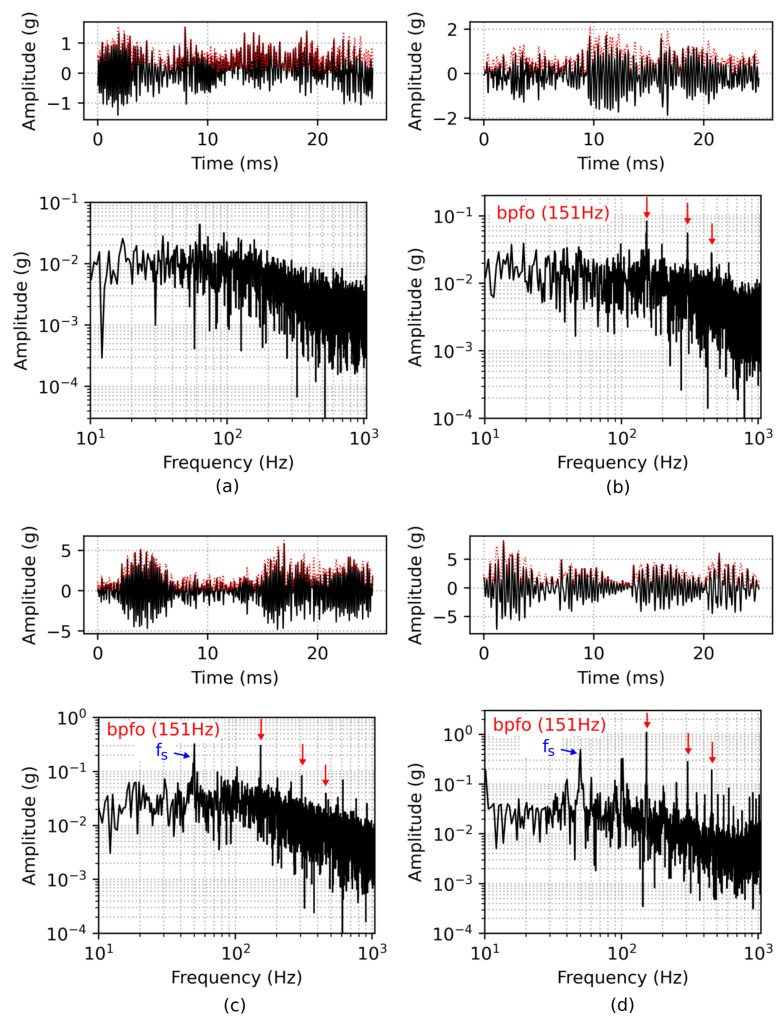
Result of envelope analysis (red dotted) from measurement and corresponding FFT. Subplot (**a**) shows the time domain and frequency domain of a healthy bearing. The vibration data and FFT of a faulty bearing with an outer ring defect size of 500 μm, 900 μm and 1200 μm is depicted in (**b**–**d**), respectively. Here, the first three harmonics are indicated as red arrows. The shaft frequency, marked as fs (50 Hz) is prominent in (**c**,**d**).

**Figure 7 sensors-22-02490-f007:**
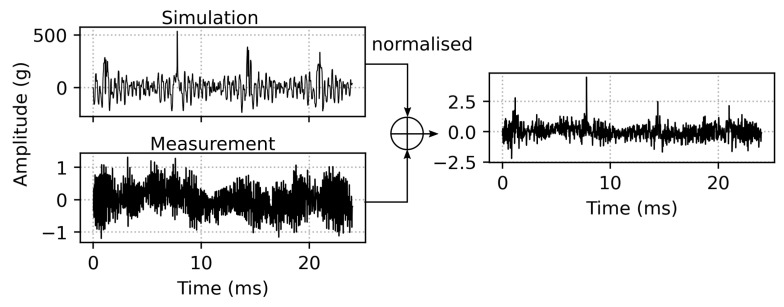
Data preprocessing steps, where the simulated time series is normalised and superimposed to the background noise obtained from the measurement of a healthy bearing.

**Figure 8 sensors-22-02490-f008:**
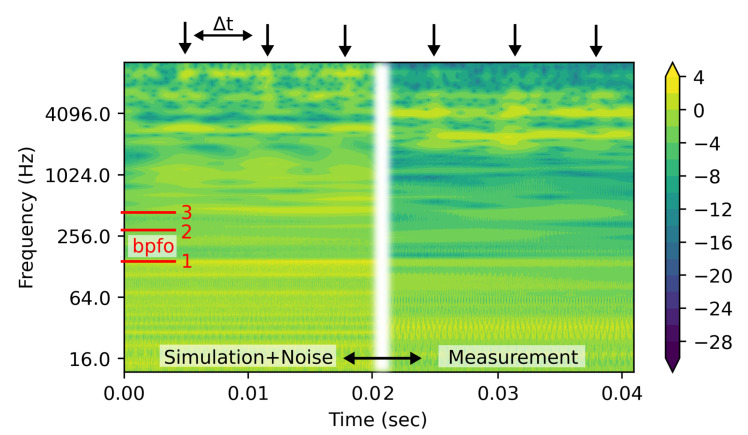
The power spectral density (PSD) plot of wavelet transformed vibration data. Data of a simulated bearing with a single outer ring fault of 500 μm and superimposed with background noise from measurement (**left**); and measurement of a faulty bearing with an outer ring defect size of 900 μm (**right**). The first three harmonics of bpfo are numbered 1–3.

**Figure 9 sensors-22-02490-f009:**
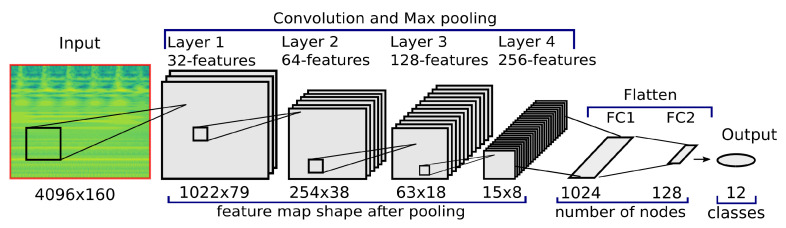
Schematics of the CNN architecture.

**Figure 10 sensors-22-02490-f010:**
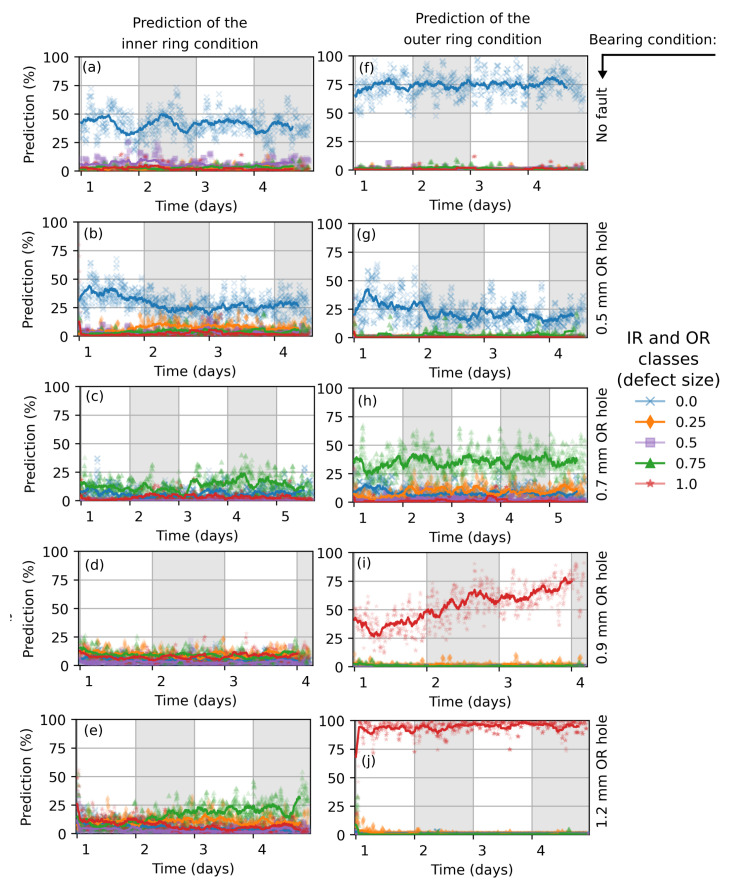
Prediction of the bearing’s condition during measurement. The CNN network was trained with 4096 × 160 images, augmented with background noise from a measured healthy bearing. The prediction of the inner- and outer ring condition is shown in plots (**a**–**e**) and (**f**–**j**), respectively. The actual bearing condition is marked on the right side (black arrow).

**Figure 11 sensors-22-02490-f011:**
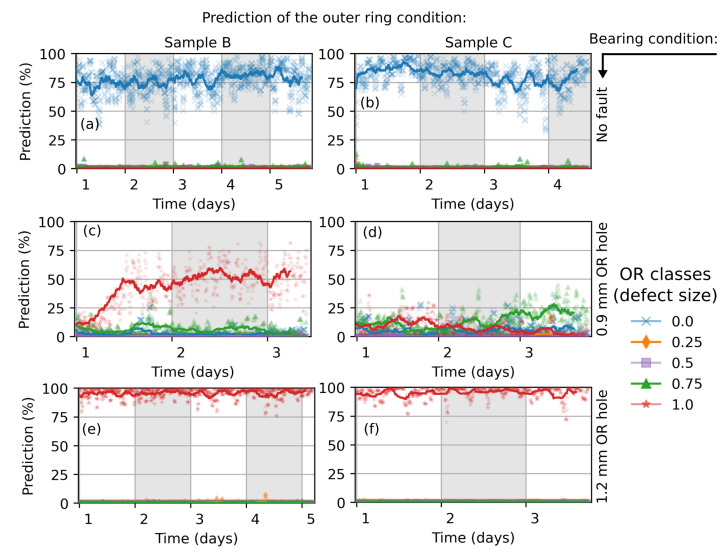
Classification of the outer ring condition of additional measured ball bearings. The CNN network was trained with 4096 × 160 images, augmented with background noise from a measured healthy bearing. The actual bearing condition is marked on the right side (black arrow).

**Figure 12 sensors-22-02490-f012:**
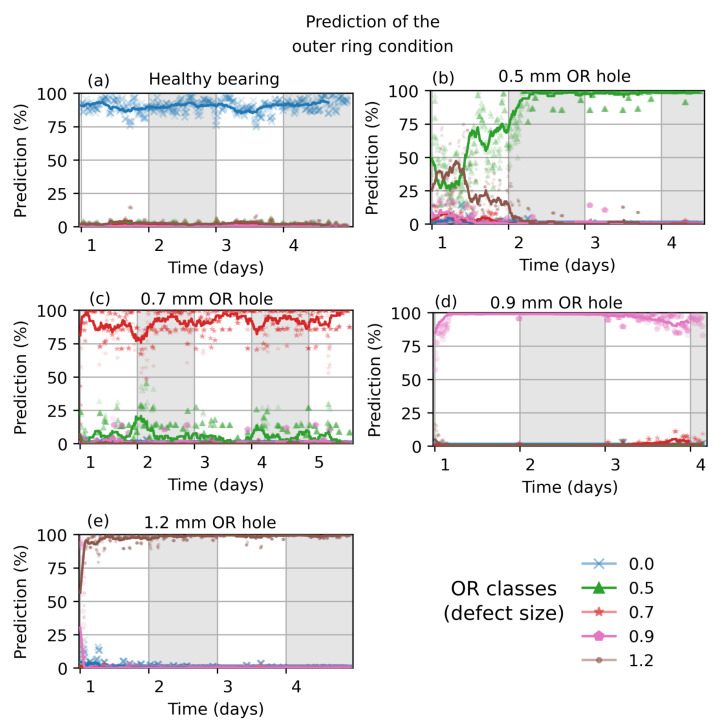
Prediction of the bearing’s outer ring condition from a network trained with measurement data. The network’s input image size is 4096 × 160.

**Figure 13 sensors-22-02490-f013:**
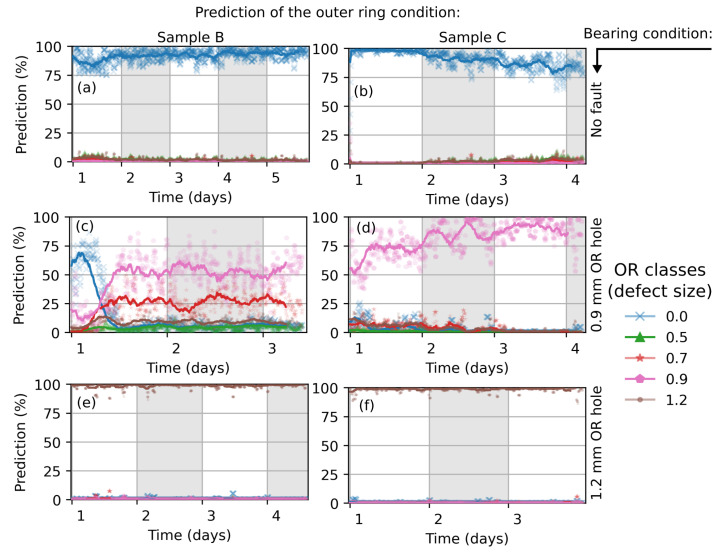
Classification of the outer ring condition of additional measured ball bearings. The actual bearing condition is marked on the right side (black arrow).

**Figure 14 sensors-22-02490-f014:**
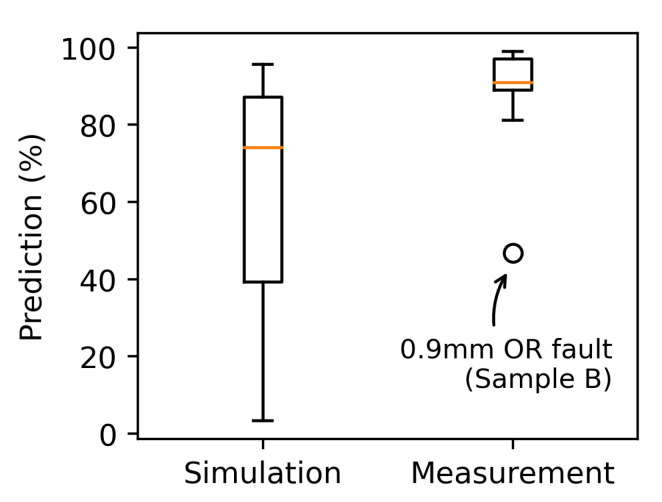
Comparison of prediction performance of CNN trained with simulation and measurement data. Only outer ring faults.

## Data Availability

The data presented in this study are available on request from the corresponding author.

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
