# Peer review of "Condition Monitoring of Ball Bearings Based on Machine Learning with Synthetically Generated Data"

_sensors, 2022, doi:10.3390/s22072490_

Round 1

Reviewer 1 Report

The paper presents an interesting, new point of view on vibration monitoring in defective mechanical bearings. The paper concerns research analysis of the rolling bearings faults' influence on the machine condition work. Authors analyse the bearing with the faults on the inner and outer race to register the vibration effect. The scientific issue concerns the use of a deep-learning convolutional neural network method for vibration analysis generated by fault mechanical components. As for the work condition of mechanical bearing, the Authors analyse the machine learning methods and present the simulation and experimental analysis effects. Moreover, the experimental validation and conclusions are proper and sufficient.

The paper is exciting and adequately prepared in research methodology. In addition, the presented scientific discussion and conclusions can attract readers' coverage.

The choice of bibliography linked to the analysed scientific problem is relevant and sufficient. Symbols (constants, variables) and functions are appropriately described and explained. Furthermore, the mathematical theory analysis accurately presents scientific and research problems and uses proper methodology.

In conclusion, I recommend the article for publication in the present form.

Author Response

Dear Reviewer,

thank you for your overwhelmingly positive feedback. You will find corresponding changes marked in the revised version of the manuscript.

Reviewer 2 Report

This work presents a simple 3D simulation model of a ball bearing under different conditions, i.e. bearing faults in the form of holes at the inner and outer race, to create a variety of vibration training data. Some comments are as follows:

  1. The challenges of rolling bearing fault diagnosis and the motivation for the proposed approach should be emphasized in the Introduction section.
  2. The authors need to highlight and summarize the contribution of this article in the introduction section.
  3. Noun abbreviations in lines 100-103 of the article need to be in uppercase letters, rather than lowercase letters.
  4. Some experimental results of the article are not clear enough after being enlarged, and the format of the pictures needs to be adjusted.
  5. The article lacks comparative experimental results to reflect the superiority of the proposed method. It is suggested to compare approach with recent ones for fault diagnosis, prognosis and predictive maintenance. For example, Prediction of remaining useful life based on bidirectional gated recurrent unit with temporal self-attention mechanism, and A review on soft sensors for monitoring, control and optimization of industrial processes.
  6. This article adopts a CNN-based method for fault diagnosis of rolling bearings. However, there is no relevant theoretical knowledge about CNN in the article, and the authors need to make detailed supplements to increase the theoretical basis of the article.
  7. The authors should add a flow diagram of the method to increase the readability of the article.
  8. The article lacks the parameter adjustment process during the CNN experiment and the corresponding hyperparameter table.

Reviewer 3 Report

Dear Authors

Congratulations to the Authors for developing their own methodology for the diagnosis of very important machine elements, such as rolling bearings. The methodology is correct because it allows clearly distinguish between healthy and defective bearings at this initial stage. There is still some work to be done by the Authors to complete the research. The problem of detecting symptoms of certain types of faults requires long-term research. Nevertheless, the Authors described in a concise and concise their simulation and testing (validation) procedure based on the currently used methods of diagnostics of rolling bearings. By the way, they also introduced their own improvements. The description of the methods used in the research and presented in the manuscript is included in the review prepared for the Editors. According to the reviewer, only one quite serious critical remark (to be fulfilled in future research) and four minor remarks are presented here. Including these minor remarks would improve the manuscript.

The most serious shortcoming in the test methodology was the assumption of a bearing load with a very low value (P = 0.1 kN) in relation to the basic dynamic load rating C = 5.4 kN of this SKF Explorer bearing. The Authors most likely suggested the fatigue load limit Pu = 0.1 kN, and this is an auxiliary parameter used to determine the life modification factor a-SKF (a-SKF = f ((eta-c) * (Pu) / P used depending on on the bearing life according to the modified SKF method According to the SKF definition: "The fatigue load limit Pu for a bearing is defined as the load level below which metal fatigue will not occur". According to calculations on the calculator https://www.skfbearingselect.com/ # / bearing-selection-start the hourly bearing life of a 6200 bearing (C = 5.4 kN) is over 2 * 10 ^ 5 hours, when the highest bearing life is practically assumed to be 5 * 10 ^ 4 hours. it follows that if the bearing rating life reaches a value greater than 100,000 hours, then there will be no fatigue damage (because such faults are included in the calculations - e.g. pitting), but there may be scuffing (scoring) due to lack of lubrication. bearing load in the test stand does not reflect the actual dynamics of the bearing in service. A copy of the calculation printout is presented below. With the same SKF calculator, you can also determine the rotational frequency and frequency of over rolling point of the bearing elements as shown in the same printout.

The second remark concerns an overly short description of the parameters of the bearing model prepared for calculations in the ADAMS system. It is mainly about not specifying the value of the clearances of the bearing balls in the cage seats to show the ball / cage interaction.

The third remark concerns the reference to the source literature [32] from 2004, which seems to suggest that formulas (1) to (4) were recently introduced in the diagnosis of bearings. Meanwhile, they were used as early as 1968, eg in [1], [2] and [3] - below attached.

The fourth remark concerns the necessity to present a clear (a bulleted or block) algorithm (bulleted or block) of the adopted actions at the beginning of the paper.

The fifth remark, or rather a recommendation, related to the lack of several important literature items in References.

The last small remark concerns the designation of the rotational velocity of the bearing in formulas (1) to (4) with the symbol rpm, and this is after all a unit of rotational velocity (moreover, the term speed is used rather in transport, and in physics - velocity). Please have a look at the referenced literature [1, 2, 3] or similar.

Yours sincerely, Reviewer.

1) Gustafsson OG, Tallian T. Detection of damage in assembled rolling element bearings. ASLE Preprint 61-AM 3B-1. 16th ASLE, Philadelphia, PA, 1961. 39pp.

2) Harris TA. Rolling bearing analysis. New York: John Wiley and Sons, 1966.

3) Broderick JJ, Burchill RF, Clark HL. Design and fabrication of prototype system for early warning of impending bearing failure. MTI Report MTI-71 TR-1 (prepared for NASA), 1972.

4) Idriss El-Thalji n , Erkki Jantunen. A summary of fault modelling and predictive health monitoring of rolling element bearings.  Mechanical Systems and Signal Processing Volumes 60–61, August 2015, Pages 252-272

5) Ying Zhang, Kangshuo Xing, Ruxue Bai, Dengyun Sun, Zong Meng. An enhanced convolutional neural network for bearing fault diagnosis based on time–frequency image. Measurement Volume 157, June 2020, 107667

6) Duy-TangHoang, Hee-JunKang: Rolling element bearing fault diagnosis using convolutional neural network and vibration image. Cognitive Systems Research. Volume 53, January 2019, Pages 42-50

7) Hui Li : Complex Morlet Wavelet Amplitude and Phase Map Based Bearing Fault Diagnosis. Proceedings of the 8th World Congress on Intelligent Control and Automation July 6-9 2010, Jinan, China 978-1-4244-6712-9/10/$26.00 ©2010 IEEE

8) C.T. Yiakopoulos and I.A. Antoniadis: Wavelet based demodulation of vibration signals generated by defects in rolling element bearings. Shock and Vibration 9 (2002) 293–306 293

9) R.X. Gao and R. Yan: Non-stationary signal processing for bearing health monitoring. Int. J. Manufacturing Research, Vol. 1, No. 1, 2006

https://www.skfbearingselect.com/#/bearing-selection-start

Calculation results

Bearing Properties6200

Designation 6200  Bearing type - Deep groove ball bearing

Principal dimensions Bore d (mm) 10, Outer diameter D (mm) 30, Width B (mm) 9

Basic load ratings Dynamic C (kN) 5.4, Static C0 (kN) 2.36

Fatigue load limit Pu (kN) 0.1

Speed ratings Limiting nllm (r/min) 36000

Minimum load    Radial Frm  (kN) 0.0192

Viscosity κ :1.81, Operating viscosity Actual  ν  (mm^2/s)  28,   Rated ν(mm^2/s)  15.4, Rated at 40°C  νref  (mm^2/s)  48.7

Lubrication condition  Viscosity ratio  κ  1.81

Bearing loads C/P :54,   Equivalent dynamic load P (kN) 0.1,   Load ratio  C/P  54

Grease life and relubrication interval tf :17100 h,   Grease quantity  Side  G(g)  1

Adjustment factors  For bearing load P  f1

Static safety factor S0 :> 20

Equivalent static load  P(kN)  0.1

Bearing rating life   Basic  L10h  (h)  > 2x10^5,   SKF   L10mh    (h)   > 2x10^5

SKF life modification factor  askf    19.58

Contamination factor  ηc    0.19

Consideration

For rating life results above 100000 hours, other failure modes than those included in the current rating life models will dominate and limit the life of the bearing. 

Frequencies

Designation 6200

Rotational frequency  Inner ring  fi   (Hz)  50,   Outer ring   fe   (Hz)  0,  Rolling element set & cage  fc

(Hz)  19.047

Rolling element about its axis  f(Hz)  99.045,  Frequency of over-rolling Point on inner ring  fip (Hz)

247.62,    Point on outer ring  fep  (Hz)  152.38,  Rolling element  frp   (Hz)  198.091

Friction M :1.62 Nmm.   END.

Round 2

Reviewer 2 Report

Well edited by the authors, I agree with the publication of this article.